# Epidemiology of Anthropometric Factors in Glioblastoma Multiforme—Literature Review

**DOI:** 10.3390/brainsci11010116

**Published:** 2021-01-16

**Authors:** Donata Simińska, Jan Korbecki, Klaudyna Kojder, Patrycja Kapczuk, Marta Fabiańska, Izabela Gutowska, Anna Machoy-Mokrzyńska, Dariusz Chlubek, Irena Baranowska-Bosiacka

**Affiliations:** 1Department of Biochemistry and Medical Chemistry, Pomeranian Medical University in Szczecin, Powstańców Wielkopolskich 72 Av., 70-111 Szczecin, Poland; d.siminska391@gmail.com (D.S.); jan.korbecki@onet.eu (J.K.); patrycja.kapczuk@pum.edu.pl (P.K.); dchlubek@pum.edu.pl (D.C.); 2Department of Anaesthesiology and Intensive Care, Pomeranian Medical University in Szczecin, Unii Lubelskiej 1 St., 71-281 Szczecin, Poland; klaudynakojder@gmail.com; 3Institute of Philosophy and Cognitive Science, University of Szczecin, Krakowska 71–79, 71-017 Szczecin, Poland; martafab98@gmail.com; 4Department of Medical Chemistry, Pomeranian Medical University in Szczecin, Powstańców Wlkp. 72 Av., 70-111 Szczecin, Poland; izagut@poczta.onet.pl; 5Department of Experimental and Clinical Pharmacology, Pomeranian Medical University in Szczecin, Powstańców Wlkp. 72 Av., 70-111 Szczecin, Poland; amachoy@pum.edu.pl

**Keywords:** glioblastoma multiforme, epidemiology, sex, gender, age, overweight, obesity, height

## Abstract

Although glioblastoma multiforme (GBM) is a widely researched cancer of the central nervous system, we still do not know its full pathophysiological mechanism and we still lack effective treatment methods as the current combination of surgery, radiotherapy, and chemotherapy does not bring about satisfactory results. The median survival time for GBM patients is only about 15 months. In this paper, we present the epidemiology of central nervous system (CNS) tumors and review the epidemiological data on GBM regarding gender, age, weight, height, and tumor location. The data indicate the possible influence of some anthropometric factors on the occurrence of GBM, especially in those who are male, elderly, overweight, and/or are taller. However, this review of single and small-size epidemiological studies should not be treated as definitive due to differences in the survey methods used. Detailed epidemiological registers could help identify the main at-risk groups which could then be used as homogenous study groups in research worldwide. Such research, with less distortion from various factors, could help identify the pathomechanisms that lead to the development of GBM.

## 1. Introduction

Glioma is a glial tumor that was first classified soon after its discovery by German pathologist Rudolf Virchow in 1865 [1]. The current World Health Organization (WHO) classification is much more complex and recommends an integrated diagnosis based on a combination of histopathological typing (that distinguishes diffuse astrocytic and oligodendroglial tumors, other astrocytic tumors, ependymal tumors, and other gliomas) and molecular typing which takes into account genetic mutations, e.g., in *TERT* (telomerase reverse transcriptase), *p53* (tumor protein P53), *IDH 1* (isocitrate dehydrogenase 1) and *IDH 2* (isocitrate dehydrogenases 2), *MGMT* (O6-alkylguanine DNA alkyltransferase), *EGFR* (epidermal growth factor receptor), and *INI1* (integrase interactor 1) genes, among other genes and molecular markers [2]. The methylation of the *MGMT* gene promoter and the chromosomal 1p/19q codeletion or mutations in the *IDH* gene are important clinical information that help not only to determine the type of cancer but also to predict the patient’s condition. A mutation in *IDH* leads to changes in the reaction catalyzed by the corresponding enzyme, resulting in the production of 2-hydroxyglutarate from α-ketoglutarate. This oncometabolite inhibits α-ketoglutarate-dependent dioxygenases [3,4]. This leads to methylation of histones and DNA 5-methylcytosine hydroxylation. Nevertheless, this effect improves the prognosis of patients with low-grade glioma and glioblastoma multiforme (GBM) [5,6]. *MGMT* promoter methylation is a prognostic factor for patients with GBM. MGMT is an enzyme that repairs damage caused by TMZ (temozolomide), [7] which means that it is responsible for the drug resistance of GBM cells to commonly used anticancer therapies. *MGMT* promoter methylation reduces the expression of *MGMT* and therefore is associated with a better prognosis for GBM patients [8]. The 1p/19q codeletion is typically found in oligodendrogliomas [2], increasing the expression of oncogens and decreasing the expression of anti-oncogens [9].

Despite the already extensive knowledge of the molecular biology of glioma, our knowledge of the pathophysiology of this cancer is still insufficient. Scientists are trying to determine the environmental and individual causes of this type of cancer. Despite an extensive body of work on various potential risk factors (ionizing radiation, alcohol intake, tobacco smoking, the use of cell phones, cytomegalovirus infection, head injuries, and drinking coffee), only ionizing radiation has been fully confirmed to be associated with the formation of glioma [3]. Another very interesting issue is the effect of drugs that nonspecifically inhibit COX (cyclooxygenase), including NSAIDs (nonsteroidal anti-inflammatory drugs) and antihistamines, on the risk of different types of histological gliomas. The researched cases were divided into 3 types: high-grade, anaplastic, and low-grade glioma. The use of NSAIDs was found to be inversely related to the occurrence of all types of glioma. The results with antihistamines were inconclusive. The tumor-promoting effect was present only in the anaplastic subtype and >10 years of drug use in the case of antihistamines. Qiu et al. [10], in their work, recalled that exacerbated COX 2 activity is associated with GBM progression, pointing to its influence on PGE2 (prostaglandin E2) and on the EP receptor (prostaglandin E receptor subtypes). In preclinical studies and in summaries of studies with patients, they confirmed the inhibitory effect of COX 2 inhibitors on tumor progression.

Among the individual factors, particular attention should be paid to the relation between resistance to allergies and immune factors and the likelihood of glioma. Linos et al., in their paper, concluded that there is strong evidence for a curious inverse relation between glioma and atopic disease. The evidence was based on information gained after searching 8 observational studies (with a total of 3450 patients with gliomas). The risk of gliomas was inversely associated with asthma, eczema, and allergy. There was no such coincidence found in the study concerning meningiomas. The authors mentioned the possible influence of the hyperactive immune system accompanying atopy on the inhibition of abnormal glioma cells. The results were unchanged via geographic conditions, study designs, and different atopic conditions. They reflect the protective effect of allergies on glioma growth. The authors pointed out that methodologic limitations are unlikely to explain the observed effect and that there is also considerable need for further prospective studies [11,12]. There are reports on the familial occurrence of glioma, and genetic predisposition to inherited glioma has been observed in 5–10% of cases [13]. Finally, some rare genetic disorders such as types 1 and 2 neurofibromatosis and tuberous sclerosis are also associated with an increased incidence of cancers including glioma [14]. An interesting issue—although rarely mentioned in the literature—is the presence of factors reducing the risk of glioma, such as the occurrence of asthma, genotypes associated with a higher risk of developing asthma, and allergies [11,12].

Equally interesting risk factors in many cancers are also simple anthropometric parameters such as gender [15,16,17], age [18,19,20], body weight [21,22], and height [23,24,25] or, additionally, in the case of central nervous system tumors, anatomical location.

To establish risk factors identifying at-risk groups that should have increased medical supervision, tests based on computed tomography (CT) or magnetic resonance imaging (MRI), which are the basic tools for the detection of gliomas, should be included [26,27,28]. This might help to increase tumor detection rates at an early stage of development, which is particularly important in light of nonspecific symptoms of malignant gliomas, often underestimated or misinterpreted by patients [26,29].

In order to determine at-risk groups and without a full understanding of the pathophysiology of glioma, it seems justified to collect extensive and detailed data on gender, age, height, weight, and histological type in the population of GBM patients. The obtained data could be helpful in creating accurate research assumptions or in faster diagnosis of nonspecific symptoms in specific groups of patients.

This work is a summary of the current data obtained from epidemiological reports available in online databases and aims to highlight the importance of maintaining accurate and reliable epidemiological records regarding gliomas. The PubMed search engine (https://pubmed.ncbi.nlm.nih.gov) was used to find articles for this work, using the following keywords: GBM and epidemiology and glioblastoma or GBM and sex or gender or age or overweight or obesity or height. From the resulting collection of articles, we selected those reviews of epidemiological data that have chapters on gender, age, and tumor location and, at the same time, have been published over the last 5 years (2015–2020). We excluded articles without specific figures as well as those not describing the histological types of tumors. In the case of height and weight, we also included older papers due their high scientific significance.

Importantly, in this paper, we present various statistical data which cannot be compared directly due to differences in definitions, methods of data collection, calculation of indicators, or delays in reporting. Nonetheless, parallel results in research may suggest the existence of certain trends regarding the effect of various GBM risk factors.

## 2. Characteristics of Glioblastoma Multiforme

Glioblastoma multiforme is a malignant primary tumor of the central nervous system. The incidence of GBM varies depending on the analyzed report, e.g., 3.20 [30], 4.06 [31], 4.17 [32], 4.40 [33], and 4.64 [34] per 100,000 inhabitants. GBM is therefore a rare disease. Alongside this, there is also a very poor prognosis [35]. Despite intensive research in recent years, the survival time has not changed significantly and, on average, does not exceed 15 months [36,37,38,39].

Highly malignant glioblastoma multiforme is divided into 3 subgroups according to the WHO classification (2016): IDH wild-type GBM (including giant cell GBM, gliosarcoma, and epithelioid GBM), IDH-mutant GBM, and GBM-NOS [2]. The molecular biomarkers of GBM are MGMT promoter methylation, *IDH* mutation, chromosome 1p/19q deletion, *TERT* promoter mutation, *TP53* mutation, *PTEN* (Phosphatase and tensin homolog) mutation, *CDKN2A* (cyclin-dependent kinase inhibitor 2A) deletion, *EGFR* amplification, *PDGFRA* (platelet-derived growth factor receptor A) amplification, *NF1* (Neurofibromin 1) mutation, *MDM2* (Mouse double minute 2 homolog) amplification, and many others [40,41]. Three molecular changes are the most crucial for diagnosing GBM, namely the concurrent gain of whole chromosome 7 and loss of whole chromosome 10 (+7/−10), *TERT* promoter mutation, and *EGFR* amplification. These help identify the tumor as GBM even though a histological test may suggest it is a low-grade tumor [42].

Additionally, we distinguish primary and secondary GBM. The former, with a de novo tumor formation, is responsible for 90% of GBM cases and affects mainly elderly patients (above 55). Only 2% of primary GBM cases show mutations in *IDH* [43]. Secondary cancer develops from astrocytic tumors or oligodendrogliomas and affects younger patients; 80% of secondary GBM cases show mutations in *IDH* [43] Secondary GBM is also often associated with the mutation of *TP53* [44].

The macroscopic and histological structures of the glioblastoma multiforme are well described by the name of the tumor itself. Macroscopically, the tumor is eminently heterogeneous, with multicolored fields of yellow necrosis, calcifications, and hemorrhagic necrosis [35,45]. Microscopically, it is also heterogeneous, characterized by pleomorphic cells: small, low-diversified, and polynuclear giants with necrosis sites, pseudopalisadium nuclei, and proliferation in the range of micronuclei [45]. In the genetic aspect, the GBM also shows its heterogeneous character (genetic multiform) with various genetic abnormalities [45] (Figure 1).

Due to the highly aggressive nature of GBM, with a wide range of complications resulting from tumor growth itself and the applied treatment—surgical, chemotherapeutic (marrow suppression, nausea), or radiotherapeutic—improving the quality of life of patients is an essential element. The treatment of GBM should always be accompanied by rehabilitation and psychological support for both the patients themselves and their relatives.

## 3. The Influence of Gender on the Incidence of Glioblastoma Multiforme

Gender has a significant impact on the functioning of the entire human organism, which results in greater susceptibility or resistance to various diseases. For example, the female gender is a risk factor for osteoporosis or cardiovascular diseases. Sex can also be a factor that determines the prognosis, choice of treatment, or type of screening. Observing gender-specific differences in the prevalence of an individual disease may help design different testing patterns in order to increase the understanding of the causes and development of particular diseases.

Figure 2 shows the GBM incidence rates in both genders presented by various authors. The rates have been standardized and calculated per 100,000 inhabitants. All presented studies show a higher prevalence of GBM in men, although due to different data collection protocols they cannot be directly compared to each other.

In order to simplify data analysis, we used the male-to-female sex ratio in the populations presented in the literature. Epidemiological reports describing the occurrence of GBM clearly indicate a higher incidence of this histological type in men (Table 1). The ratio ranged from 1.2 to 2.6 for studies conducted on adult populations, 1.34–2.5 for general populations, and 1–1.5 for children. The increased incidence of GMB in men is associated with differences in the activity of retinoblastoma protein and functioning of cancer stem cells [46].

Gittleman et al. [33] described a statistically significant increase in the incidence of GBM in men, and according to Walker et al. [31], it is unlikely that the continual predominance of men as patients with malignant histological types of glioma is accidental. Additionally, there are also studies that report the same gender distribution among GBM patients of different geographic origin [36,62]. On this basis, it could be concluded that gender and the resulting biological differences are associated with a higher incidence of GBM among men. This is contrary to the conclusions of Tian et al. [54], who show statistically significant differences in gender distribution between races. There are some studies that also show statistically significant gender-related differences in GBM survival [52,54], although there are others where the differences are not statistically significant [33,34,36]. In the study by Shabihkhani et al. [52], the male gender was found to have a statistically significant risk factor with regard to survival, with a hazard ratio equal to 1.053 (CI (confidence interval) 1.022–1.085), while Tian et al. [54] and Xie et al. [51] indicated a higher survival of female patients with GBM at hazard ratios equal to 0.906 (CI 0.859–0.954) and 0.937 (CI 0.914–0.961), respectively. Gender does not influence the incidence of IDH1 mutation in GBM [63,64]. One study indicates increased TERT promoter mutation in men with GBM [65], although this has not been confirmed in other reports [66,67,68].

## 4. Effect of Age on the Incidence of Glioblastoma Multiforme

Age, a factor analyzed in research on many diseases, is also a contributing factor of most cancers, with recommendations for specific screening tests for patients of certain ages. Therefore, it seems obvious that age can also significantly influence the incidence of GBM (Figure 3).

In the study by Hansen et al., the majority of patients with GBM (83%) were over 50 years old [38], while Cheo et al. and Gosh et al. reported that it was most common in those in their 60s [53,57]. In the study by Li et al., 47.9% of patients were aged 65+ years, with incidences peaking between 75 and 79 years of age, followed by a drop [48].

In children, the frequency of tumors in a central location decreases with age (*p* < 0.0001) [39]. In addition, an increase in age in the entire population correlates with a decrease in survival rates [38] and is considered a significant risk factor [37,55], with a hazard ratio of 1.54 (CI 1.46–1.62, *p* < 0.00001) [55]. In a study by Bohn et al. [36], the hazard ratio for 3-year survival in GBM was 2.18 (1.91–2.49) times higher for patients over 50 years of age (*p* < 0.001).

Age as a significant mortality risk factor was also confirmed in children aged from 6 to 10 years, with a hazard ratio equaling 1.408 (CI: 1.069–1.854, *p* = 0.01), and in those aged from 11 to 19 years, with a hazard ratio equal to 1.406 (CI: 1.094–1.806, *p* = 0.0077) [25]. According to Ghosh et al., the median survival rate is highest between the ages of 31 and 40 [57]: 8.8 months for patients under 50 years of age and 4.55 months for those older than 50 (*p* = 0.016). Other reports confirm that the median survival rate of GBM patients decreases with age [36,57,69], and according to Cheo et al., it is the lowest among those aged 70 and above. Importantly, the patient’s age is associated with IDH mutation. IDH1 mutation is more frequent in younger patients than in older patients [63,64,70], while the opposite is true for TERT promoter mutation [65,66,67,68], PTEN mutation, and MGMT methylation [70]. In young patients with GBM (18–45 years), TP53 mutation is more frequent than in older patients (46 years and above) [70].

## 5. Glioblastoma Multiforme Incidence Depending on Tumor Location in the Central Nervous System

Most epidemiological reports show GBM tumors in the central nervous system, with the detailed location determined using various methods. Although some works refer to the ICD-O-3 Code (International Statistical Classification of Diseases and Related Health Problems), which facilitates a reliable reading of the results, this practice is not common enough to apply it to every epidemiological report. Some works described the location only in a very general way, e.g., by indicating a hemisphere.

According to the study by Fabbro-Peray et al. [32] on 2053 GBM patients in France, 47.1% had a tumor located in the right hemisphere, 45.7% had tumors in the left hemisphere, and 7.2% showed a “central” location. In the study of Ghosh et al., on Indian patients [57], the incidence of a tumor in the right hemisphere was found in 50.81% of cases, in the left hemisphere was found in 39.34%, while in the central location was found in 9.83% of cases. Of course, it should not be concluded that the Indian population is more likely to have GBM in the right hemisphere than the French population, as the latter study was conducted on a considerably smaller group of 61 patients. In addition, a question also arises whether this location description is accurate enough to be used for interstudy comparisons. Another classification adds the category of “other locations”. For example, in the population of 1173 patients in the U.S. aged under 19 years studied by Liu et al. [39], 48.02% of tumors were located in the hemispheres of the brain, 30.16% were in the “brain center”, and 22.83% were in another location. Another study, also in the U.S., showed that 72% of the tumors had a supratentorial location and that only 1.1% were subtentorial, with 29.9% showing a different location [48]. The prevalence of GBM in a supratentorial location (71.6%) was confirmed by a study by Bin Abdulrahman et al. [50]. A study on children by Lam et al. [61] showed that supratentorial tumors accounted for 61.9% of cases and were associated with better survival than subtentorial tumors (*p* = 0.002).

A common way to describe tumor location is the SEER (Surveillance, Epidemiology, and End Results) system with the following categories: “not a paired site”, when the origin can be attributed to neither side; “right: origin of primary”; “left: origin of primary”; “only one side involved”, when the tumor is unilateral with unknown origin; “bilateral involvement”, used for a bilateral tumor where the side of origin is unknown; and “paired site”, when the tumor has a central site of origin. The SEER system was used in the studies of Xu et al. [49] and Xie et al. [51], with “not a paired site” reported in 16.1% and 16% of cases, “paired site” reported in 1.3% and 0.9%, “one side involvement” reported in 81.3% and 81.6%, and “bilateral involvement” reported in 1.3% and 1.5% of cases, respectively. Both studies were conducted by Chinese teams and covered the same time period, albeit Xie et al.’s [51] analysis covered a period that was two years longer, focusing on the relationship between civil status and survival and including data on the location of the tumor.

The most common way to present location data is to specify the following structures: frontal lobe, temporal lobe, parietal lobe, occipital lobe, cerebellum, brain stem, and other location. Although the percentage distribution of the location in this classification varies between reports (Table 2), one of the recurring trends is the rare occurrence of GBM in the cerebellum, although the work of Shieh et al. [71] contradicts this by indicating that there is a 25% share of this location. Another correlation is the rare occurrence of GBM in the occipital lobe among all brain lobes. A study by Li et al. [72] compared the incidence of GBM in the frontal and temporal lobes in 406 Chinese patients and showed that the tumor location closer to the ventricle is statistically significantly more frequent in the frontal than the temporal lobe. Additionally, the lobes differed in the IDH mutation (*p* = 0.021) or MGMT promoter methylation status (*p* = 0.012). Patients with a tumor in the frontal lobe showed significantly shorter survival levels than patients with a tumor in the temporal lobe. However, no difference was observed in the size or sides of the tumor in both lobes. Patients with a tumor crossing the midline and with a central location had worse survival outcomes than others [38].

## 6. Influence of Overweight and Obesity on the Incidence of Glioblastoma Multiforme

It is estimated that, in 2015, 1.9 billion adults were affected by overweight and another 609 million were obese [73]. In the U.S. alone, 64% of the population is overweight and 28% is obese. The problem of abnormal body weight mainly affects the populations of developed countries, but it is also growing in developing countries [73]. Obesity is associated with many diseases, including cardiovascular diseases, hypertension, and type II diabetes. In recent years, obesity has become also a risk factor for many cancers, including endometrial cancer, kidney cancer, and esophageal adenocarcinoma [74]. Obesity also increases cancer mortality [75]. However, on the other hand, obesity has been shown to reduce the risk of some cancers, including lung cancer [21] and prostate cancer [22]. 

Studies available on PubMed show that obesity does not affect the risk of morbidity or mortality among GBM patients (Table 3). The first published study in 1989, which considered more than 11,000 patients from Norway, did not show a link between BMI and GBM [76]. This was confirmed by later studies. For example, Jones et al. did not find any correlation between BMI and the overall survival of GBM patients [77]. Wiedmann et al. did not show a correlation between BMI and the risk of GBM and other gliomas [78]. However, it seems that abnormal body weight does affect the processes associated with GBM development. It has been proven that abnormally high BMI at the age of 18 increases the incidence of this type of cancer at a later age [79]. This was confirmed by Little et al. on a group of 1111 patients with glioma, including 694 patients with GBM [80]. Being underweight at 21 years of age was associated with a decrease in the incidence of glioma, and obesity (BMI > 30) at 21 years of age increased the incidence of glioma at a later age—each additional kilogram of weight in the overweight increased the risk of glioma by 4% [80]. At the time of diagnosis, BMI was not associated with the incidence of glioma or GBM. Nevertheless, BMI may serve as a predictive factor. Potharaju et al. showed, in a study on 392 GBM patients that overweight and obesity improved overall survival [81]. On the other hand, a study by Jones et al. on a 3 times larger population of GBM patients showed that BMI does not affect overall survival levels [77].

## 7. Height as a Risk Factor for GBM Incidence

Height is a risk factor for many cancers. Meta-analyses have shown that it can be associated with colorectal cancer [23], kidney cancer [24], pancreatic cancer [25], prostate cancer [82], and thyroid cancer [24]. This is related to the fact that genes and signaling pathways responsible for high growth also participate in cancer processes [83]. Another important link between height and cancer risk is the level of various hormones responsible for being tall. In particular, cancer incidence is promoted by high levels of insulin-like growth factors (IGF) and growth hormones (GH) [84,85,86] that naturally occur in tall people.

Available studies also show that growth is strongly linked to the incidence of GBM (Table 4), although genetic research has not shown that height has any influence on the frequency of IDH1 mutation in glioma [78]. The first published study, from 1989, shows that each additional 15 cm of height in men is associated with a 36% higher incidence rate of GBM [77]. In women, on the other hand, it was only 18% higher and was not statistically significant. The same correlation was observed by Wiedmann et al., who observed that, in a group of 3102 patients with GBM, both taller men and women were at a higher risk of GBM [78], with specific numbers similar to those reported by Helseth et al. [76]. In another study on a group of 321 GBM patients, Cote et al. [87] observed that every additional inch of height increased the risk of GBM by 7%. Similar results were obtained by Kitahara et al., who showed that, in both sexes, height is a significant risk factor for GBM [88], although much smaller than in the three aforementioned studies. In men, each additional 5 cm of height resulted in an 8% higher risk of GBM, while in women, it was only 4%.

## 8. Conclusions

Differences in the results from different epidemiological reports may be caused by differences in intrapopulation gender ratios; insufficient sample sizes in relation to population size; different definitions of variables, e.g., in terms of including gliosarcoma or large cell glioma as GBM; or differences in the formulated inclusion and exclusion criteria. Due to the above reasons, these reports should not be directly compared with each other. The analysis of the data collected in this article seems to indicate a higher incidence of GBM in men or an increasing risk of its occurrence with age. Finally, our review indicates the need for more accurate and precise reporting of epidemiological data on the location of GBM tumors.

The use of molecular tests on GBM tumors provides a more accurate determination of the prevalence of each GBM subgroup. The differences in how often these subgroups occur between various studied populations must be compared against environmental conditions. Individual GBM subtypes differ in the initial stages of development due to differences in the molecular and immunological characteristics of the patient as well as in other external factors. The survival curves presented in various reports do not indicate any significant trends for each subgroup. Identical aggressive surgical resections followed by radio- and chemotherapy have resulted in either very long or extremely short survival times, suggesting that only the natural characteristics of the tumor, when analyzed along with that of the host’s, can provide a reasonable prognosis. The fact that this type of tumor occurs relatively frequently with significantly different survival rates led us to review the available literature data. This review shows that further comparative research is required in this field to obtain more standardized information that could be used to increase the efficacy of and to modify the process of the treatment.

## Figures and Tables

**Figure 1 brainsci-11-00116-f001:**
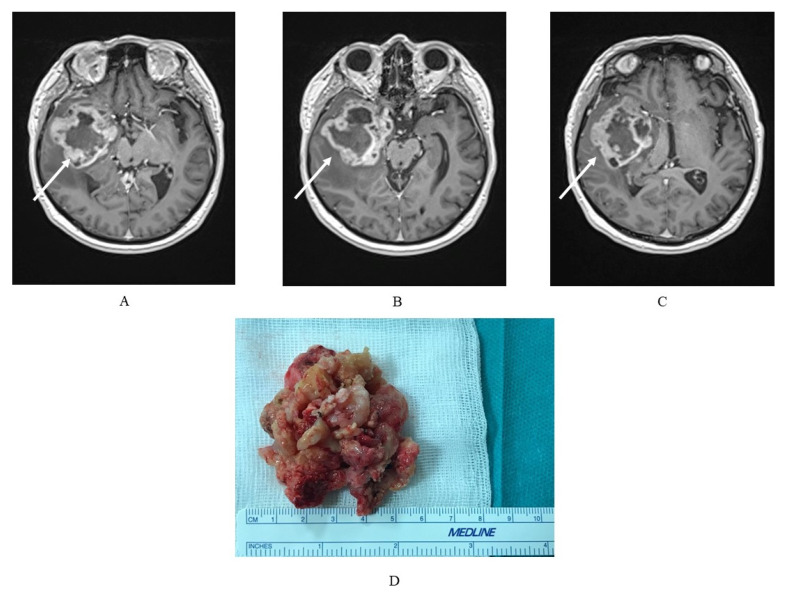
(**A**–**C**) Magnetic resonance imaging (MRI) exam of a glioblastoma multiforme (GBM) in a 75-year-old patient with a tumor localized in the right temporal lobe with a heterogeneous signal, and, characteristically for GBM, strong, irregular, marginal contrast enhancement and (**D**) the GBM tumor mass of the same patient, where macroscopically a visible, heterogeneous structure is typical.

**Figure 2 brainsci-11-00116-f002:**
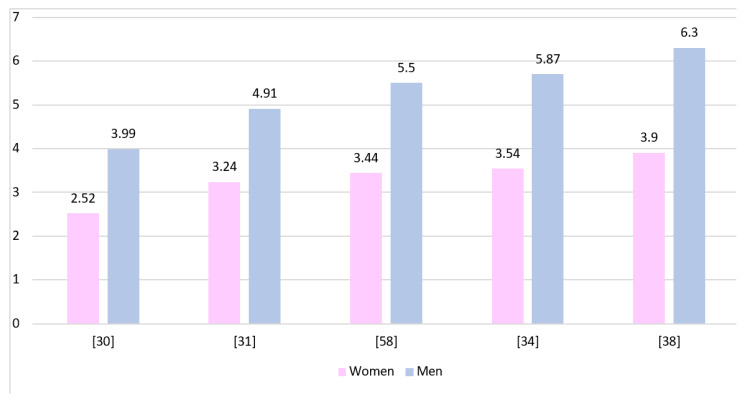
The incidence rates of GBM presented for both genders and calculated per 100,000 inhabitants of each population.

**Figure 3 brainsci-11-00116-f003:**
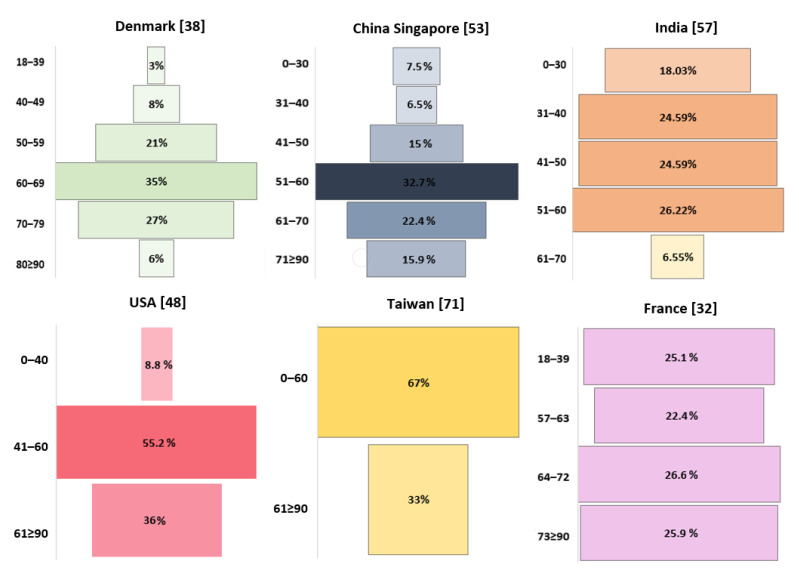
Age-related incidence of GBM.

**Table 1 brainsci-11-00116-t001:** Male-to-female sex ratio in studies on GBM.

Country	Dominant Sex	Gender Ratio	Years Range	Population Size	Patient Age	Literature
Adults						
USA	Men	1.148	1997–2010	3759	>66	Burton et al., 2015 [47]
USA	Men	1.347	1973–2014	49,124	>20	K. Li et al., 2018 [48]
USA	Men	1.35	2004–2013	24,262	All	Xu et al., 2017 [49]
USA	Men	1.364	1985–2014	25,117	18–60+	Bin Abdulrahman et al., 2019 [50]
USA	Men	1.375	2000–2014	6,430,706	>20	Gittleman et al., 2018 [33]
USA	Men	1.38	2004–2015	30,767	18–70+	Xie et al., 2018 [51]
USA	Men	1.395	2010–2014	33,473	>18	Bohn et al., 2018 [36]
USA	Men	1.45	2001–2011	1826	>19	Shabihkhani et al., 2017 [52]
Denmark	Men	1.545	2009–2014	1364	>18	Hansen et al., 2018 [38]
China Singapore	Men	1.547	2002–2011	107	13–85	Cheo et al., 2017 [53]
USA	Men	1.596	2000–2008	6586	18–70	Tian et al., 2018 [54]
Sweden	Men	1.6	2001–2012	143	18–99	Bruhn et al., 2018 [37]
Germany	Men	1.629	2011–2014	2382	>18	De Witt Hamer et al., 2019 [55]
China Hong Kong	Men	1.833	2003–2005 2010–2012	68	>18	Chan et al., 2017 [56]
India	Men	2.588	2012–2014	61	15–68	Ghosh et al., 2017 [57]
Adults and children						
Spain	Men	1.338	1993–2014	463	0–74+	Fuentes-Raspall et al., 2017 [58]
USA	Men	1.359	2010–2014	56,421	All	Ostrom et al., 2017 [30]
Canada	Men	1.459	2009–2013	5830	All	Walker et al., 2019 [31]
France	Men	1.5	2008–2015	2053	All	Fabbro-Peray et al., 2019 [32]
England	Men	1.503	2007–2011	10,743	0–85+	Brodbelt et al., 2015 [34]
Iran	Men	2.476	2011–2016	73	All	Salehpour et al., 2019 [59]
Children						
USA	Men	1.102	1973–2013	154	0–18	Maxwell et al., 2018 [60]
USA	Women	1.049	1973–2013	252	0–5	Liu et al., 2018 [39]
	Men	1.278	1973–2013	287	6–10	
	Men	1.438	1973–2013	634	11–19	
USA	Men	1.455	2000–2010	302	<20	Lam et al., 2018[61]

Gender ratio—the quantitative ratio of the dominant sex to the other sex in the study group.

**Table 2 brainsci-11-00116-t002:** Summary of percentage distributions of GBM locations in the central nervous.

Country	Frontal Lobe	Temporal Lobe	Parietal Lobe	Occipital Lobe	Cerebellum	Central	Brainstem	Brain	NOS Chamber	NOS Brain	Overlapping Brain Damage	Other	Years Range	Population Size	Literature
USA	38	35	21	6	-	-	-	-	-	-	-	-	2010–2014	33,473	Bohn et al., 2018 [36]
England	24.9	21.8	16.7	4.8	0.5	-	0.4	-	-	-	-	30.9	2007–2011	10,743	Brodbelt et al., 2015 [34]
USA	25.25	23.6	16.7	4.2	0.5	-	0.5	3.6	0.4	0.4	17.6	-	2000–2008	6586	Tian et al., 2018 [54]
Denmark	30	32	19	11	-	8	-	-	-	-	-	-	2009–2014	1364	Hansen et al., 2018 [38]
USA	22.8	22.8	54.3	-	-	-	-	-	-	-	-	-	2000–2010	302	Lam et al., 2018 [61]
India	42.62	9.83	9.83	4.91	-	-	-	-	-	-	-	31.78	2012–2014	61	Ghosh et al., 2017 [57]
Taiwan	31	13	6	9	25	-	-	-	-	-	-	17	2005–2016	44	Shieh et al., 2020 [71]

**Table 3 brainsci-11-00116-t003:** Summary of studies on the association of BMI with the risk of GBM.

Country	Tested Rate	Observations	Population Size	Literature
Norway	Risk factor	No significant interactions	11,144	Helseth et al., 1989 [76]
Norway	Risk factor	No significant interactions	3102	Wiedmann et al., 2017 [78]
USA	Risk factor	BMI at 18 correlates with the incidence rate at a later age	480	Moore et al., 2009 [79]
USA	Risk factor	BMI at 21 correlates with the incidence rate at a later age	1111 glioma (group with 694 GBM cases)	Little et al., 2013 [80]
USA	Overall survival	No significant interactions	1259	Jones et al., 2010 [77]
India	Prognostic Marker/overall survival	Obesity/overweight improves prognosis	392	Potharaju et al., 2018 [81]

**Table 4 brainsci-11-00116-t004:** Height as a GBM risk factor: review of the literature.

Country	Tested Rate	OR	Observations	Population Size	Literature
Norway	Risk factor	1.36 (+15 cm)	Positive correlation in men	6391 (men)	Helseth et al., 1989 [76]
Norway	Risk factor	1.18 (+15 cm)	No statistically significant correlation	4752 (women)	Helseth et al., 1989 [76]
USA, Finland, Australia, Sweden	Risk factor	1.08 (+5 cm)	Positive correlation in men	405 (men)	Kitahara et al., 2012 [88]
USA, Finland, Australia, Sweden	Risk factor	1.04 (+5 cm)	Positive correlation in women	308 (women)	Kitahara et al., 2012 [88]
Norway	Risk factor	1.24 (+10 cm)	Positive corelation	3102	Wiedmann et al., 2017 [78]
USA	Risk factor	1.07 (+1 inch)	Positive correlation	321	Cote et al., 2018 [87]

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
