# Peer review of "Epidemiology of Anthropometric Factors in Glioblastoma Multiforme—Literature Review"

_brainsci, 2021, doi:10.3390/brainsci11010116_

Round 1
Reviewer 1 Report
The authors summarize clinical studies on glioblastomas epidemiologically and implicate an anthropometic influence on occurrence. They suggest standardized epidemiological registers for the identification of risk groups. They further propose for risk groups to go under increased medical supervision, including screening tests in order to increase early stage-detection rates.
Classification based on molecular characteristics should include general tests suggested by WHO for IDH mutation, 1p/19q codeletion, and H3F3A mutations next to specific IDH1/IDH2, ATRX, TERT, TP53, PTEN mutations, NF1 deletion, EGFR amplification, PDGFRA amplification, MGMT methylation status a.o. [Chen et al. Neurotherapeutics. 2017].
GBM have been previously suggested to hold a higher prevalence for men in case of WT-IDH status according to WHO report [Lai et al 2011 J Clin Oncol., Nobusawa et al. 2009 Clin Cancer Res., Bleeker et al. 2010 Acta Neuropathol J., Sun et al. 2014 The J Clin Investigation.]
The Figure showing age-related incidences among various nationalities could be presented by using similar age groups for better comparison.
If data available, a comparison of molecular markers among identified risk groups would be beneficial.
A comparison of Kaplan Meier curves throughout studies according to clinical features would round the summary out.
Author Response
Rev1.
The authors summarize clinical studies on glioblastomas epidemiologically and implicate an anthropometic influence on occurrence. They suggest standardized epidemiological registers for the identification of risk groups. They further propose for risk groups to go under increased medical supervision, including screening tests in order to increase early stage-detection rates.
Classification based on molecular characteristics should include general tests suggested by WHO for IDH mutation, 1p/19q codeletion, and H3F3A mutations next to specific IDH1/IDH2, ATRX, TERT, TP53, PTEN mutations, NF1 deletion, EGFR amplification, PDGFRA amplification, MGMT methylation status a.o. [Chen et al. Neurotherapeutics. 2017].
According to the reviewer's guidelines, the classification has been improved.
GBM have been previously suggested to hold a higher prevalence for men in case of WT-IDH status according to WHO report [Lai et al 2011 J Clin Oncol., Nobusawa et al. 2009 Clin Cancer Res., Bleeker et al. 2010 Acta Neuropathol J., Sun et al. 2014 The J Clin Investigation.]
The manuscript has been completed as recommended by the reviewer.
The Figure showing age-related incidences among various nationalities could be presented by using similar age groups for better comparison.
Data for the figures were taken from various publications. We do not have raw data. For this reason we are not able to change the the figures.
If data available, a comparison of molecular markers among identified risk groups would be beneficial.
The incidence of molecular markers in these GBM patient groups has been added.
A comparison of Kaplan Meier curves throughout studies according to clinical features would round the summary out.
Reviewer 2 Report
38-39: The WHO 2016 is the current classification and changes have been made compared to the 2007 including removal of the oligoastrocytic part. A new WHO classification is expected to be out soon withing 12 months.
40-43: More important genetics markers are currently incorporated into classifications including TERT, loss of 7, gain of 10, etc. Not sure if EGF means EGFR?
43-44: Sentence seems somewhat awkward. Important why? And why neurosurgeons rather than oncologists?
52: is there a list of abbreviations included in the paper? GBM, IDG, MGMT, etc.
52-59: reference? Is this specific to high grade gliomas?
68: Recommend using the term glioma rather than GBM based on the current literature and new/expected newer classification as for example we are soon going to rename IDH mutant GBM to IDH mutant high grade glioma. Also this makes more sense when talking about genetic risk factors as most increase the risk of glioma in general including low grade rather than only high grade.
73-78:
- “on glioma tumor incidence” not sure about incidence here. On glioma patient population sounds better.
- Why only gender, age, height, weight, and histological type? Is this based on previous literature? Also why histological types if you are looking for risk factors retrospectively on one population of patient?
- I understand this is an introduction but justification of the study design/question is essential for the reader.
78-79: Not sure bout the sections of the study. I was expecting methodology after this. Could you please elaborate?
88-90: Are we looking at all CNS tumors gender ratio? The studies are heterogenous, some of these studies included gliomas only, some neuro-epithelial tumors, some all CNS tumors. Did these studies include also metastases? We know certain tumors are more gender specific e.g oligodendrogliomas are more common in males. I think if you are looking at population study you should include one group.
91-102: Same issue as above, very different groups. It is well known that meningioma is more common in females as a group for example but you cannot compare for example meningioma grade I with for example either glioma grade III. I would avoid using the term “cancer”. The following graph is wrong as many CNS tumors are graded differently and some don’t have grade IV. This should be explained in this part as for example meningioma are the most common tumor and does not have a grade IV and thus would bias the data toward lower grades.
109-118: This part is confused as I’m not sure to which classification you are referring to; 2007 and 2016? Did all the studies use the same classification?
122-130: Needs to be reviewed, again which classification? Older studies used 2007 and newer used 2016. Did you incorporate both? Can you compare studies based on different classification?
137-156: Theme here is for GBM or high grade glioma. Since this is a literature review, I think we it’s important to mention the most recent trend in the classification based on the available literature; https://academic.oup.com/nop/advance-article/doi/10.1093/nop/npaa055/5899025
I think the paper is interesting but needs major updates and uniformity of population study, adding methodology part explaining why you selected only the referenced paper, adding conclusion part, etc. The conclusion part is ambiguous. I think age is well known risk factor. You did not comment about acquired risk factors for different population and focused on gender. Then you talked about survival and treatment which is a very big and complicated topic and I don’t think that suffice.
Author Response
Rev 2.
38-39: The WHO 2016 is the current classification and changes have been made compared to the 2007 including removal of the oligoastrocytic part. A new WHO classification is expected to be out soon withing 12 months.
It has been corrected according to the reviewer's comment.
40-43: More important genetics markers are currently incorporated into classifications including TERT, loss of 7, gain of 10, etc. Not sure if EGF means EGFR?
The genetic marker TERT has been included, of course EGF should be EGFR.
43-44: Sentence seems somewhat awkward. Important why? And why neurosurgeons rather than oncologists?
The sentence has been changed and no longer includes a specific medical specialty.
52: is there a list of abbreviations included in the paper? GBM, IDG, MGMT, etc.
The paper has no list of abbreviations (as it's not required by the journal) but now each abbreviations is explained when it appears for the first time in the paper.
52-59: reference? Is this specific to high grade gliomas?
The list of symptoms is given for malignant gliomas, which is further specified in the text.
68: Recommend using the term glioma rather than GBM based on the current literature and new/expected newer classification as for example we are soon going to rename IDH mutant GBM to IDH mutant high grade glioma. Also this makes more sense when talking about genetic risk factors as most increase the risk of glioma in general including low grade rather than only high grade.
The nomenclature has been changed according to the reviewer's comments.
73-78:
“on glioma tumor incidence” not sure about incidence here. On glioma patient population sounds better
The sentence has been changed according to the reviewer's comments.
Why only gender, age, height, weight, and histological type? Is this based on previous literature? Also why histological types if you are looking for risk factors retrospectively on one population of patient?
The factors were selected based on criteria repeated in epidemiological reports. However, detailed reporting in terms of histological types may also help highlight the differences in the individual risk factors. This paper is based on data collected before the introduction of the current classification. In addition this fragment refers to all types of CNS tumors, so the histological type seems to be important here.
I understand this is an introduction but justification of the study design/question is essential for the reader.
The justification has been added.
78-79: Not sure bout the sections of the study. I was expecting methodology after this. Could you please elaborate?
The section on methodology is brief, but it is a review paper, not a meta-analysis with a specific algorithm used for article selection.
88-90: Are we looking at all CNS tumors gender ratio? The studies are heterogenous, some of these studies included gliomas only, some neuro-epithelial tumors, some all CNS tumors. Did these studies include also metastases? We know certain tumors are more gender specific e.g oligodendrogliomas are more common in males. I think if you are looking at population study you should include one group.
All quoted reports except 14 and 15 describe primary cancers or tumors of the central nervous system. Papers 14 and 15 do describe more detailed groups and have therefore been deleted from the table.
91-102: Same issue as above, very different groups. It is well known that meningioma is more common in females as a group for example but you cannot compare for example meningioma grade I with for example either glioma grade III. I would avoid using the term “cancer”. The following graph is wrong as many CNS tumors are graded differently and some don’t have grade IV. This should be explained in this part as for example meningioma are the most common tumor and does not have a grade IV and thus would bias the data toward lower grades.
Due to the misleading nature of the fragment, we decided to remove it and the corresponding figure from the paper.
109-118: This part is confused as I’m not sure to which classification you are referring to; 2007 and 2016? Did all the studies use the same classification?
All studies quoted in this excerpt analyze data from patients diagnosed before 2016, and so they are based on the old classification from 2007.
122-130: Needs to be reviewed, again which classification? Older studies used 2007 and newer used 2016. Did you incorporate both? Can you compare studies based on different classification?
The studies on which the fragment is based analyze data from the years 1994-2013 and 2010-2014, so they use the classification from 2007 or even older, but the data was standardized by the authors of those articles. Due to the mentioned discrepancies, at the beginning of the article we mention possible differences in classification between the reports, and therefore we point out the need for more accurate and precise reporting of epidemiological data in the future.
137-156: Theme here is for GBM or high grade glioma. Since this is a literature review, I think we it’s important to mention the most recent trend in the classification based on the available literature; https://academic.oup.com/nop/advance-article/doi/10.1093/nop/npaa055/58990258-39
We have added information from the suggested literature.
Reviewer 3 Report
The authors provide an excellent literature review on epidemiologic studies on anthropometric factors influencing the incidence of gliomas and especially glioblastomas. This topic is highly important as a base to identify risk factors and therefore understand the pathophysiology of these tumors. It also has practical use for the future definition of homogeneous subgroups for clinical studies. As more and more molecular data becomes available for clinical routine treatment, the division of brain tumors into subgroups will more and more important. These subgroups might possibly be mainly defined by molecular markers, however the impact of anthropomorphic factors will be important as well. Therefore the topic is relevant and interesting for many of your readers. The manuscript is indeed highly original, as it gives a good overview on the topic. The authors have included and discussed all relevant studies on the topic. It therefore adds to the already published material, as these studies are discussed critically. The authors found differences in the various epidemiological reports existing. The deduction from these differences is that these reports cannot be compared directly to each other and future investigations should use the same definitions of variables and the same inclusion and exclusion criteria. Also the location of gliomas should be reported more accurately and precisely. All relevant literature is cited. The conclusions drawn by the authors are convincing and consistent with the evidence and arguments presented. The paper is well written, the text is clear and easy to read. Also, the linguistic style is very good, which makes the manuscript easy to read and understand.
Author Response
Thank you very much for very thorough review.
Round 2
Reviewer 2 Report
44-45: Would add among other genes and molecular markers.
46: Would elaborate more on why MGMT and why 1p/19q important or just delete the whole sentence. IDH for example is also crucial. MGMT is not used for classification but rather a prognostic marker.
47-49: Could you reference this part please. The CT or MRI part.
55-62: Needs a reference.
67-68: allergies and immune factors are interesting and evolving, I think further elaboration would be nice. How strong is the evidence on it being a risk factor? Or does it affect survival?
68-69: I think its more than just reports, 5-10% of all glioma is not uncommon in my opinion.
72-75: I think this pertains to the 67-68 part, would lump together and maybe elaborate more.
75: Is the COX-2 proposed preventative effect immunological? Is the mechanism know or hypothesized? Would elaborate more on this too.
76-78: You did not mention anything so far regarding age, gender, height, weight, or histological types and their association to neoplasm but then you say: “Due to the inability to determine at-risk groups and without a full understanding of the pathophysiology of glioma, it seems justified to collect extensive and detailed data on gender, age, height, weight and histological type in the population of GBM patients.”. Not sure why you decided to chose these factors age, height, etc. and not what you mentioned earlier being a potential risk factors i.e immunological and COX-2. What's your question and or hypothesis on the former risk factors?
88: Again, I think the methodology is not clear. What search term you used? which website/database? which one did you include and which one did you exclude? etc.
94: The table and the following part include studies of all CNS tumors not only gliomas. This study from the title and introduction is about high grade glioma. Not sure which population are we looking at, if all CNS tumor then the whole introduction part should be redone and the title should be changed. If glioma only, then in my opinion we should include only glioma studies and there are many. Additionally, there are many more population studies e.g France: https://pubmed.ncbi.nlm.nih.gov/9070498, Denmark: https://pubmed.ncbi.nlm.nih.gov/28861666/, and many others.
Author Response
44-45: Would add among other genes and molecular markers.
The statement has been added in this passage.
46: Would elaborate more on why MGMT and why 1p/19q important or just delete the whole sentence. IDH for example is also crucial. MGMT is not used for classification but rather a prognostic marker.
The aim of this passage was to emphasize the role of molecular markers not only in tumor classification, but also in forecasting the patient's condition. The sentence has been expanded to make it more understandable and new literature items have been added to describe this aspect in more detail.
47-49: Could you reference this part please. The CT or MRI part.
The sentence has been transformed and references have been added.
55-62: Needs a reference.
The passage about the glidee symptoms has been removed in order to preserve the consistency of the text.
67-68: allergies and immune factors are interesting and evolving, I think further elaboration would be nice. How strong is the evidence on it being a risk factor? Or does it affect survival?
The fragment has been extended
68-69: I think its more than just reports, 5-10% of all glioma is not uncommon in my opinion.
The sentence has been changed.
72-75: I think this pertains to the 67-68 part, would lump together and maybe elaborate more.
The paragraphs have been merged.
75: Is the COX-2 proposed preventative effect immunological? Is the mechanism know or hypothesized? Would elaborate more on this too.
The fragment has been extended.
76-78: You did not mention anything so far regarding age, gender, height, weight, or histological types and their association to neoplasm but then you say: “Due to the inability to determine at-risk groups and without a full understanding of the pathophysiology of glioma, it seems justified to collect extensive and detailed data on gender, age, height, weight and histological type in the population of GBM patients.”. Not sure why you decided to chose these factors age, height, etc. and not what you mentioned earlier being a potential risk factors i.e immunological and COX-2. What's your question and or hypothesis on the former risk factors?
The work aims at a collective analysis of the factors popularly reported in epidemiological reports. The occurrence of allergies is not commonly published in these reports, but due to the importance of this issue we decided to discuss it in the introduction. Additionally, we have added a fragment that explains the need for a deeper analysis of the factors described in the article.
88: Again, I think the methodology is not clear. What search term you used? which website/database? which one did you include and which one did you exclude? etc.
The methodology is now described in more detail in the article.
94: The table and the following part include studies of all CNS tumors not only gliomas. This study from the title and introduction is about high grade glioma. Not sure which population are we looking at, if all CNS tumor then the whole introduction part should be redone and the title should be changed. If glioma only, then in my opinion we should include only glioma studies and there are many. Additionally, there are many more population studies e.g France: https://pubmed.ncbi.nlm.nih.gov/9070498, Denmark: https://pubmed.ncbi.nlm.nih.gov/28861666/, and many others.
This chapter was written to outline the general epidemiology of CNS tumors and to draw the reader's attention to the need to use narrower groups for analysis. However, we do agree with the reviewer and we decided to delete this part of the article.